# Semi-Supervised End-To-End Contrastive Learning For Time Series Classification

## Abstract

Time series classification is a critical task in various domains, such as finance, healthcare, and sensor data analysis. Unsupervised contrastive learning has garnered significant interest in learning effective representations from time series data with limited labels. The prevalent approach in existing contrastive learning methods consists of two separate stages: pre-training the encoder on unlabeled datasets and fine-tuning the well-trained model on a small-scale labeled dataset. However, such two-stage approaches suffer from several shortcomings, such as the inability of unsupervised pre-training contrastive loss to directly affect downstream fine-tuning classifiers, and the lack of exploiting the classification loss which is guided by valuable ground truth. In this paper, we propose an end-to-end model called SLOTS (Semi-supervised Learning fOr Time clasSification). SLOTS receives semi-labeled datasets, comprising a large number of unlabeled samples and a small proportion of labeled samples, and maps them to an embedding space through an encoder. We calculate not only the unsupervised contrastive loss but also measure the supervised contrastive loss on the samples with ground truth. The learned embeddings are fed into a classifier, and the classification loss is calculated using the available true labels. The unsupervised, supervised contrastive losses, and classification loss are jointly used to optimize the encoder and classifier. We evaluate SLOTS by comparing it with ten state-of-the-art methods across five datasets. On an EEG-based emotion recognition task using the DEAP dataset with only 10% labeled data, SLOTS significantly outperforms two-stage baselines, achieving up to a 16.10% higher F1 score (compared to TS-TCC) and a 38.49% higher absolute accuracy (compared to TS2Vec) when the labeling ratio increases to 100%. SLOTS also attains the best performance on four diverse datasets with an average 3.55% margin in F1. In various evaluation setups, including leave-trials-out and leave-subjects-out, SLOTS consistently achieves top performance. The results demonstrate that SLOTS is a simple yet effective framework. When compared to the two-stage framework, our end-to-end SLOTS utilizes the same input data, consumes a similar computational cost, but delivers significantly improved performance. Crucially, our end-to-end framework is model-agnostic, allowing it to be seamlessly integrated with any existing self-supervised contrastive model in order to enhance its performance. We release code and datasets at `https://anonymous.4open.science/r/SLOTS-242E`.

## 1 Introduction

Time series data, a sequence of data points collected at regular intervals over time Li (2020), is prevalent across various fields such as economics, meteorology, healthcare, and transportation Harutyunyan et al. (2019); Rezaei & Liu (2019); Ravuri et al. (2021); Sezer et al. (2020); Su & Wen (2022); Deng et al. (2021). However, time series analysis often faces limitations of the scarcity of data labels. Factors such as domain expertise requirements, extensive manual efforts, and privacy concerns constrain the acquisition of high-quality data annotations Mohsenvand et al. (2020). Unsupervised (or self-supervised) contrastive learning has emerged as a significant and effective technique for learning valuable representations from time series data without relying on labels Wang et al. (2023). This approach operates by prompting the model to generate similar representations for distinct views or

Figure 1: (a) Illustration of a standard self-supervised contrastive model for classification tasks. The whole framework includes two stages: pre-training and fine-tuning. (b) Proposed end-to-end framework. Our model exhibits two properties that are unattainable in conventional two-stage frameworks: 1) the unsupervised contrastive loss, calculated on unlabeled data, contributes to the optimization of the classifier, enabling more effective utilization of the unlabeled data; 2) the incorporation of a newly added supervised contrastive loss further enhances performance and facilitates more efficient learning, resulting in a more powerful and robust model.

augmentations of the same data instance, while simultaneously driving the representations of different instances further apart Pöppelbaum et al. (2022); Chen et al. (2020); Jiang et al. (2021).

The majority of existing unsupervised contrastive learning methods follow a two-stage paradigm Kan et al. (2022); Shen et al. (2022), as illustrated in Figure 1 (a). First, during the pre-training stage, an unlabeled dataset is fed into an encoder that maps the input data sample to a latent space. In this space, a contrastive loss is calculated based on instance discrimination. The unsupervised contrastive loss is then used to optimize the encoder Li et al. (2022a); Lee et al. (2022); Li et al. (2022d). Subsequently, in the fine-tuning stage, a small-scale labeled dataset is fed into the pre-trained model to generate representations. A supervised classification loss is calculated to optimize the classifier and/or fine-tune the encoder.

While the two-stage contrastive learning approach has demonstrated effectiveness in various domains, it also presents challenges that may hinder better performance. One notable shortcoming of this process is that the unsupervised pre-training contrastive loss cannot directly influence downstream fine-tuning classifiers, potentially limiting the optimization of the final model. Furthermore, this approach fails to fully exploit the classification loss, which is guided by valuable ground truth information. As a result, the potential benefits of incorporating such information during the learning process are not fully realized in the traditional two-stage contrastive learning framework. Moreover, considering the two stages as a whole, it becomes evident that the self-supervised learning framework actually receives a semi-supervised dataset (integrating unlabeled samples in pretraining and labeled samples in fine-tuning). Then, a natural idea is to directly design a semi-supervised end-to-end model to fully exploit the semi-labeled data.

To address these questions, we propose an end-to-end contrastive framework named SLOTS (Semi-supervised Learning fOr Time clasSification), as shown in Fig. 1 (b). The end-to-end model allows for the simultaneous training of the encoder and classifier, unifying model optimization, reducing intermediate computational resources, and learning task-specific features. SLOTS takes a semi-labeled dataset as input, which is identical to that of a conventional two-stage framework. Compared to a traditional two-stage framework, our SLOTS: 1) receives the same dataset (partially unlabeled and partially labeled samples); 2) has the same model components (an encoder and a classifier); and 3) calculates not only the unsupervised contrastive loss and supervised classification loss but also introduces a supervised contrastive loss, computed with available true labels (see ablation study).

Regarding loss functions, unsupervised contrastive loss learns representations by maximizing agreement between differently augmented views of the same unlabeled sample. Supervised contrastive learning brings samples with same categories closer to each other using a supervised contrastive loss in the embedding space. To jointly train the end-to-end framework, we develop a hybrid loss that weighted aggregates unsupervised contrastive loss, supervised contrastive loss, and classification loss.

We note the trade-off between adaptability and model performance. While two-stage models may be more effective in transfer learning or situations where pre-training and fine-tuning involve different datasets, our SLOTS model is a superior choice for achieving optimal performance on a specific dataset (Section 5; Appendix E). We evaluate the SLOTS model on five time series datasets covering a large set of variations: different numbers of subjects (from 15 to 38,803), different scenarios

(neurological healthcare, human activity recognition, physical status monitoring, etc.), and diverse types of signals (EEG, acceleration, and vibration, etc.). We compare SLOTS to ten state-of-the-art baselines. Results show that SLOTS outperforms the two-stage baselines with an improvement of up to 16.10% in the F1 score with only 10% labeled data on the emotion recognition dataset DEAP Koelstra et al. (2012). Furthermore, SLOTS achieves the highest performance with substantial margins (3.55% on average in F1 score) compared with the strongest baselines on broad tasks.

## 2 RELATED WORK

**Self-supervised contrastive learning with time series.** Self-supervised learning, popular for learning intrinsic information from unlabeled data, has been adapted for contrastive learning in the computer vision domain Misra et al. (2016); Mikolov et al. (2013); Devlin et al. (2019); Le-Khac et al. (2020); He et al. (2020); Chen et al. (2020). Some works have applied contrastive learning to time series Shen et al. (2022); Banville et al. (2021); Li et al. (2022c), such as TS2Vec Yue et al. (2022), TS-TCC Eldele et al. (2021), and data augmentation schemes with mixing components Wickstrøm et al. (2022). Zhang et al. introduced a self-supervised pre-training strategy for time series, modeling Time-Frequency Consistency (TF-C) Zhang et al. (2022a), while Khosla et al. extended self-supervised methods to fully supervised approaches, leveraging emotion label information in EEG samples Khosla et al. (2020).

**Semi-supervised contrastive learning** Semi-supervised learning has been integrated with contrastive learning for its ability to train networks with limited labeled data and large-scale unlabeled data Arazo et al. (2020); Miyato et al. (2019); Sajjadi et al. (2016). Yang et al. proposed CCSSL to improve pseudo-label quality Yang et al. (2022); Singh et al. presented a temporal contrastive learning (TCL) framework for semi-supervised action recognition Singh et al. (2021) and later proposed a simple Contrastive Learning framework for semi-supervised Domain Adaptation (CLDA) Singh (2021); Kim et al. introduced SelfMatch, combining contrastive self-supervised learning and consistency regularization Kim et al. (2021); Inoue et al. proposed a semi-supervised contrastive learning framework based on generalized contrastive loss (GCL), unifying supervised metric learning and unsupervised contrastive learning Inoue & Goto (2020).

**End-to-end contrastive framework for time series.** In contrastive learning, including self-supervised and the majority of semi-supervised methods, the two-stage framework is mainstream, involving a self-supervised pre-training stage followed by a supervised fine-tuning stage Eldele et al. (2021); Xiao et al. (2021); Li & Metsis (2022); Kostas et al. (2021). In many studies, when researchers refer to "semi-supervised contrastive" learning, they often mean that the overall two-stage framework utilizes both labeled and unlabeled samples. Although the entire model is considered semi-supervised, the pre-training stage remains unsupervised, as it primarily focuses on learning representations from unlabeled data. The two-stage framework has certain limitations (Section 1), which has led us to explore end-to-end solutions. Notably, there are several pioneering works in developing end-to-end models within the context of contrastive learning. Li et al. proposed an EEG-based emotion recognition method based on an efficient end-to-end CNN and contrastive learning (ECNN-C) Li et al. (2022a). They evaluated their models on two common EEG emotion datasets, DEAP Koelstra et al. (2012) and DREAMER Katsigiannis & Ramzan (2018), demonstrating that the end-to-end network achieves high accuracy in time series classification. Furthermore, Zhang et al. Zhang et al. (2022b) introduced a novel training strategy called Semi-supervised Contrastive Learning (SsCL), which combines the well-known contrastive loss in self-supervised learning with the cross-entropy loss in semi-supervised learning.

Nevertheless, ECNN-C solely relies on supervised learning, which requires a sufficient number of labeled samples and lacks plug-in compatibility. On the other hand, SsCL focuses on leveraging pseudo-labels and seeks an optimal approach to identify suitable positive pairs among all samples. Since the input data and problem definitions differ from ours, we do not directly compare our method with theirs in the experiments.

In this paper, we present a novel semi-supervised framework that achieves optimal performance with minimal labeled samples and can be seamlessly integrated into various architectures. Our approach

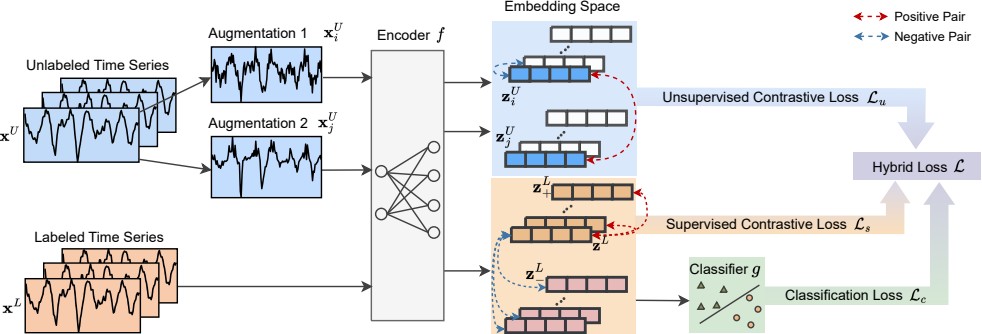

Figure 2: Overview of the proposed SLOTS. Our end-to-end model accepts a semi-labeled dataset consisting of unlabeled time series and labeled time series samples. We perform augmentations on the unlabeled sample $\boldsymbol{x}^{\mathrm{U}}$, generating two views $\boldsymbol{x}_i^{\mathrm{U}}$ and $\boldsymbol{x}_j^{\mathrm{U}}$. These two views are fed into an encoder $f$ that maps samples to a latent embedding space, where we compute the unsupervised contrastive loss $\mathcal{L}_u$ through instance discrimination. Concurrently, we evaluate the supervised loss $\mathcal{L}s$ on the representations learned from the labeled sample $\boldsymbol{x}^{\mathrm{L}}$. The $\boldsymbol{z}_+^{\mathrm{L}}$ associates with the same label as $\boldsymbol{z}^{\mathrm{L}}$ while $\boldsymbol{z}_-^{\mathrm{L}}$ belongs to a different label. Additionally, we calculate the classification loss $\mathcal{L}_c$ on the labeled samples. The model is optimized using a hybrid loss $\mathcal{L}$.

systematically combines unsupervised contrastive loss, supervised contrastive loss, and classification loss to jointly update the model, maximizing the utilization of information from the data.

## 3 METHOD

### 3.1 PROBLEM FORMULATION

Suppose a semi-labelled time series dataset $\mathcal{D}$ contains two parts: the unlabeled subset $\mathcal{D}^{\mathrm{U}}$ with $N$ unlabeled sample $\boldsymbol{x}^{\mathrm{U}}$, and the labeled subset $\mathcal{D}^{\mathrm{L}}$ with with $M$ labeled sample $\boldsymbol{x}^{\mathrm{L}}$ which associates with a label $y \in \{1, \dots, C\}$. The $C$ denotes the number of classes. We denote the set of all available labels in $\mathcal{D}$ as $\mathcal{Y}$. Without loss of generality, in the following descriptions, we focus on univariate (single-channel) time series $\boldsymbol{x}^{\mathrm{U}}$ or $\boldsymbol{x}^{\mathrm{L}}$, while noting that our approach can accommodate multivariate time series of varying lengths across datasets.

Let $f$ be the encoder that maps a time series sample to its representation in the latent space, and let $g$ be the classifier that maps the latent representation to the class probabilities. **We aim to** learn the optimal encoder $f^*$ and classifier $g^*$ by minimizing the following hybrid loss:

$$(f^*, g^*) = \arg\min_{f,g} \left[ \lambda_1 \mathcal{L}_u(f, \mathcal{D}^{\mathrm{U}}) + \lambda_2 \mathcal{L}_s(f, \mathcal{D}^{\mathrm{L}}, \mathcal{Y}) + \lambda_3 \mathcal{L}_c(f, g, \mathcal{D}^{\mathrm{L}}, \mathcal{Y}) \right], \quad (1)$$

where $\mathcal{L}_u(f, \mathcal{D}^{\mathrm{U}})$ is the unsupervised contrastive loss, $\mathcal{L}_s(f, \mathcal{D}^{\mathrm{L}}, \mathcal{Y})$ is the supervised contrastive loss, $\mathcal{L}_c(f, g, \mathcal{D}^{\mathrm{L}}, \mathcal{Y})$ is the classification loss, and $\lambda_1$, $\lambda_2$, and $\lambda_3$ are the hyperparameters that balance the contributions of the three losses. The successful $f^*$ and $g^*$ can accurately predict the label for a new time series sample.

### 3.2 MODEL ARCHITECTURE

We present the model pipeline of the proposed SLOTS in Figure 2. SLOTS receives a semi-labeled dataset $\mathcal{D}$, including an unlabeled subset $\mathcal{D}^{\mathrm{U}}$ and a labeled subset $\mathcal{D}^{\mathrm{L}}$.

For each unlabeled sample $\boldsymbol{x}^{\mathrm{U}} \in \mathcal{D}^{\mathrm{U}}$, SLOTS first generates two augmented views $\boldsymbol{x}_i^{\mathrm{U}}$ and $\boldsymbol{x}_j^{\mathrm{U}}$ using data augmentation techniques, such as jittering Eldele et al. (2021) and timestamp masking Yue et al. (2022). We use the subscripts $_i$ and $_j$ to mark the augmented samples. Then, these two augmented views are passed through the encoder $f(\cdot)$, resulting in their corresponding latent representations:

$$\boldsymbol{z}_i^{\mathrm{U}} = f(\boldsymbol{x}_i^{\mathrm{U}}), \quad \boldsymbol{z}_j^{\mathrm{U}} = f(\boldsymbol{x}_j^{\mathrm{U}}). \quad (2)$$

We calculate the unsupervised contrastive loss $\mathcal{L}_u(f, \mathcal{D}^{\mathrm{U}})$ in this embedding space. The $\boldsymbol{z}_i^{\mathrm{U}}$ and $\boldsymbol{z}_j^{\mathrm{U}}$ constructs a positive pair as they originated from the same input sample $\boldsymbol{x}^{\mathrm{U}}$; in contrast, the

embedding derived from different samples are regarded as negative pairs. Our preliminary results suggest a project header is not necessary in SLOTS.

For each labeled time series $\boldsymbol{x}^{\text{L}} \in \mathcal{D}^{\text{L}}$, SLOTS directly sends it to the encoder $f$ to learn the representation $\boldsymbol{z}^{\text{L}}$ through:

$$\boldsymbol{z}^{\text{L}} = f(\boldsymbol{x}^{\text{L}}). \tag{3}$$

The unlabeled sample $\boldsymbol{x}^{\text{U}}$ and labeled sample $\boldsymbol{x}^{\text{L}}$ are from the same original signal space and are mapped by the same encoder $f$. Therefore, we assume that the embeddings $\boldsymbol{z}_i^{\text{U}}$, $\boldsymbol{z}_j^{\text{U}}$, and $\boldsymbol{z}^{\text{L}}$ are located in the same latent space. In this space, we measure the supervised loss $\mathcal{L}_s(f, \mathcal{D}^{\text{L}}, \mathcal{Y})$. We assume that samples associated with the same label form positive pairs, while others form negative pairs. See more details of loss functions in Section 3.3.

Next, the representation $\boldsymbol{z}^{\text{L}}$ is fed into a classifier $g$, which produces the estimated sample label $\hat{y} = g(\boldsymbol{z}^{\text{L}})$. With the ground truth $y \in \mathcal{Y}$, we measure the classification loss $\mathcal{L}_c(f, g, \mathcal{D}^{\text{L}}, \mathcal{Y})$.

### 3.3 SEMI-SUPERVISED CONTRASTIVE LOSS

**Unsupervised contrastive loss.** The unsupervised contrastive loss $\mathcal{L}_u(f, \mathcal{D}^{\text{U}})$ aims to encourage the model to learn meaningful latent representations by minimizing the distance between the embeddings of augmented views of the same unlabeled sample, while maximizing the distance between the embeddings of different samples in the latent space. We leverage the contrastive learning framework and adopt the normalized temperature-scaled cross-entropy loss Chen et al. (2020) for this purpose.

For each unlabeled sample $\boldsymbol{x}^{\text{U}} \in \mathcal{D}^{\text{U}}$, we generate two augmented views $\boldsymbol{x}_i^{\text{U}}$ and $\boldsymbol{x}_j^{\text{U}}$. After passing them through the encoder $f$, we obtain their corresponding embeddings $\boldsymbol{z}_i^{\text{U}}$ and $\boldsymbol{z}_j^{\text{U}}$. The unsupervised contrastive loss, averaging across all samples, is computed as:

$$\mathcal{L}_u(f, \mathcal{D}^{\text{U}}) = \frac{1}{N} \sum_{i=1}^{N} -\log \frac{\exp(\text{sim}(\boldsymbol{z}_i^{\text{U}}, \boldsymbol{z}_j^{\text{U}})/\tau)}{\sum_{k=1, k \neq i}^{N} \exp(\text{sim}(\boldsymbol{z}_i^{\text{U}}, \boldsymbol{z}_k^{\text{U}})/\tau)}, \tag{4}$$

where sim is a similarity function (e.g., cosine similarity), $\tau$ is a temperature hyperparameter. By minimizing the unsupervised contrastive loss, the model learns to generate similar representations for augmented views of the same sample, while generating dissimilar representations for different samples in the latent space, which helps the model learn useful features from the unlabeled data.

**Supervised contrastive loss.** The supervised contrastive loss $\mathcal{L}_s(f, \mathcal{D}^{\text{L}}, \mathcal{Y})$ forces the model to learn discriminative features for labeled samples by maximizing the similarity between the representations of samples that belong to the same class while minimizing the similarity between the representations of samples from different classes. Compared to unsupervised loss, this is stronger guidance because true sample labels are directly involved as supervision.

Given a labeled time series $\boldsymbol{x}^{\text{L}} \in \mathcal{D}^{\text{L}}$ with its corresponding ground truth label $y \in \mathcal{Y}$. We measure the supervised contrastive loss based on representation $\boldsymbol{z}^{\text{L}}$:

$$\mathcal{L}_s(f, \mathcal{D}^{\text{L}}, \mathcal{Y}) = -\log \frac{\sum_{y=y_+} \exp(\text{sim}(\boldsymbol{z}^{\text{L}}, \boldsymbol{z}_+^{\text{L}})/\tau)}{\sum_{y \neq y_-} \exp(\text{sim}(\boldsymbol{z}^{\text{L}}, \boldsymbol{z}_-^{\text{L}})/\tau)}. \tag{5}$$

The $\boldsymbol{z}_+^{\text{L}}$ is a representation derived from a sample that belongs to the same class as $\boldsymbol{z}^{\text{L}}$, i.e., $y = y_+$. Conversely, $\boldsymbol{z}_-^{\text{L}}$ is derived from a sample with a different label, such that $y \neq y_-$. Minimizing the supervised contrastive loss $\mathcal{L}_s(f, \mathcal{D}^{\text{L}}, \mathcal{Y})$ enables the model to produce more closely related representations for labeled samples belonging to the same class, while generating more distinct representations for labeled samples from different classes. This process promotes the learning of discriminative features that are beneficial for the classification task.

**Classification loss.** We incorporate a classification loss to directly optimize the model toward the classification task. We use the softmax cross-entropy loss, which is a widely used loss function for multi-class classification problems.

Given a labeled sample $\boldsymbol{x}^{\text{L}}$, we obtain its predicted class probabilities $\hat{\boldsymbol{y}} = g(f(\boldsymbol{x}^{\text{L}}))$. Let $\hat{y}_c$ denote the predicted probability of class $c \in \{1, \cdots, C\}$ for a given labeled sample. We calculate the

classification loss between predicted class probabilities $\hat{y}$ and the ground-truth labels $y$:

$$\mathcal{L}_c(f, g, \mathcal{D}^{\mathrm{L}}, \mathcal{Y}) = -\frac{1}{M} \sum_{i=1}^{M} \sum_{c=1}^{C} \mathbb{1}(y_i = c) \log \hat{y}_c, \qquad (6)$$

where $\mathbb{1}(y_i = c)$ is an indicator function that equals 1 when the true label of the $i$-th labeled sample is $c$, and 0 otherwise. This loss directly optimizes the model for predicting the correct class labels.

**Hybrid loss.** The proposed end-to-end SLOTS is trained by jointly minimizing a hybrid loss $\mathcal{L}$, which is a weighted sum of the unsupervised contrastive loss, supervised contrastive loss, and classification loss, as described in the problem formulation (Equation 1). The three loss functions complement each other in terms of improving the classification results. During training, we update the encoder $f$ and classifier $g$ by minimizing the hybrid loss. This enables the model to learn effective representations from both unlabeled and labeled samples, while simultaneously improving its classification performance by directly optimizing the predicted class probabilities. The end-to-end training process allows the model to adaptively balance the contributions of the unsupervised, supervised, and classification losses to achieve the best performance on the given task.

## 4 EXPERIMENTS

**Datasets.** (1) **DEAP**, a multimodal dataset based on the stimulation of music video materials, is used to analyze human emotional states Koelstra et al. (2012). It contains 32 subjects monitored by 32-channel EEG signals while watching 40 minutes (each minute is a trail) of videos. (2) **SEED** includes 62-channel EEG data of 15 subjects when they are watching 15 film clips with three types of emotions Zheng et al. (2019). (3) **EPILEPSY** monitors the brain activities of 500 subjects with a single-channel EEG sensor (174 Hz) Andrzejak et al. (2001). Each sample is labeled in binary based on whether the subject has epilepsy or not. (4) **HAR** has 10,299 9-dimension samples from 6 daily activities Anguita et al. (2013). (5) **P19** (PhysioNet Sepsis Early Prediction Challenge 2019) includes 38,803 patients that are monitored by 34 sensors in ICU Reyna et al. (2020). More dataset details are in Appendix A.

**Baselines.** We evaluate our SLOTS model by comparing it with ten baselines, consisting of five state-of-the-art contrastive methods that employ a two-stage framework and their improved end-to-end variants. The five state-of-the-art methods include TS2vec Yue et al. (2022), Mixing-up Wickstrøm et al. (2022), TS-TCC Eldele et al. (2021), SimCLR Chen et al. (2020), and TFC Zhang et al. (2022a). To ensure a fair comparison, we create a two-stage version of SLOTS, named *SLOTS (two-stage)*, which can be directly compared with the five basic baselines. In addition, to adapt these baselines to our end-to-end framework, we modify the baselines and denote the updated versions with a '++', resulting in a total of ten baselines. For instance, the end-to-end version of TS2Vec is referred to as *TS2Vec++*. In this paper, **unless explicitly stated otherwise, SLOTS refers to the *SLOTS (end-to-end)* model.**

**Implementation.** We adopt a dilated convolutional block as backbone for encoder $f$ following Yue et al. (2022). A dilated module with three blocks is designed based on separable convolution. Each block consists of four distinct convolutional layers. The first layer employs a 2D convolution with a $1 \times 1$ kernel. The second and third layers utilize kernels of size $1 \times 3$ and $3 \times 1$, respectively, replacing the $3 \times 3$ kernel in a manner similar to the MobileNet model. This reduces the number of parameters by 33%Li et al. (2022b), enabling the network to operate more efficiently. The fourth layer is a spatial convolution with a depth multiplier of 2, doubling the number of channels from the first deep convolution layer. This layer allows the model to have sufficient channels to employ a reduction factor in bottleneck structuresFreeman et al. (2018). The classifier model $g$ utilizes a linear fully connected layer with a single hidden unit as its structure. We employ temporal masking for time series augmentations. See more details of experimental settings (e.g., hardware, baseline details, and hyper-parameters) in Appendix B.

### 4.1 COMPARING FRAMEWORKS: END-TO-END VERSUS TWO-STAGE

Table 1 presents the performance on the DEAP dataset in a leave-trials-out setting with labeling ratios ranging from 10% to 80%. Each baseline with the '++' suffix is trained in an end-to-end framework. Here, we provide a detailed analysis of the results in terms of the absolute values of the

Table 1: Performance comparison on DEAP dataset while the label ratio (the proportion of labeled samples in the training dataset) ranging from 10% to 100%.

| Ratio | Models | Accuracy | Precision | Recall | F1 Score | AUROC | AUPRC |
|---|---|---|---|---|---|---|---|
| 10% | TS2Vec | 0.5125±0.0149 | 0.5020±0.0158 | 0.5029±0.2001 | 0.5002±0.0196 | 0.5200±0.0059 | 0.5365±0.0067 |
| | TS2Vec++ | 0.5218±0.0123 | 0.5109±0.0214 | 0.5100±0.0211 | 0.5057±0.0236 | 0.5447±0.0146 | 0.5590±0.0125 |
| | SimCLR | 0.5150±0.0123 | 0.5173±0.0145 | 0.5154±0.0169 | 0.5096±0.0169 | 0.5194±0.0156 | 0.5292±0.0154 |
| | SimCLR++ | 0.5862±0.0415 | 0.5912±0.0472 | 0.5848±0.0352 | 0.5716±0.0316 | 0.5874±0.0312 | 0.5815±0.0296 |
| | TFC | 0.5006±0.0152 | 0.5167±0.0254 | 0.5255±0.0445 | 0.5249±0.0445 | 0.5099±0.0365 | 0.5102±0.0421 |
| | TFC++ | 0.6605±0.0547 | 0.6549±0.0365 | 0.6533±0.0542 | 0.6504±0.0654 | 0.7069±0.0412 | 0.7084±0.0357 |
| | TS-TCC | 0.5025±0.0631 | 0.4977±0.0614 | 0.4992±0.0748 | 0.4946±0.0861 | 0.5344±0.0743 | 0.5485±0.0763 |
| | TS-TCC++ | 0.5547±0.0367 | 0.5431±0.0378 | 0.5409±0.0369 | 0.5358±0.0412 | 0.5575±0.0347 | 0.5680±0.0360 |
| | MixingUp | 0.5250±0.0165 | 0.5357±0.0162 | 0.5326±0.0198 | 0.5075±0.0186 | 0.5447±0.0164 | 0.5478±0.0157 |
| | MixingUp++ | 0.6037±0.0974 | 0.5806±0.0874 | 0.6025±0.0962 | 0.5463±0.0871 | 0.6704±0.0634 | 0.6713±0.0631 |
| | SLOTS (two-stage) | 0.6276±0.0541 | 0.6393±0.0495 | 0.6138±0.0547 | 0.6055±0.0563 | 0.6704±0.0471 | 0.6480±0.0396 |
| | **SLOTS (end-to-end)** | **0.6636±0.0763** | **0.6598±0.0791** | **0.6571± 0.0773** | **0.6556±0.0781** | **0.7115±0.0995** | **0.7118±0.0953** |
| 20% | TS2Vec | 0.5250±0.0205 | 0.5397±0.0231 | 0.5308±0.0264 | 0.5004±0.0213 | 0.5361±0.0064 | 0.5452±0.0063 |
| | TS2Vec++ | 0.5754±0.0096 | 0.6076±0.0068 | 0.5876±0.0067 | 0.5346±0.0099 | 0.6470±0.0037 | 0.6517±0.0048 |
| | SimCLR | 0.5225±0.0223 | 0.5199±0.0248 | 0.5168±0.0296 | 0.5128±0.0284 | 0.5235±0.0198 | 0.5412±0.0196 |
| | SimCLR++ | 0.6061±0.0521 | 0.6078±0.0543 | 0.5928±0.0469 | 0.5855±0.0437 | 0.6376±0.0379 | 0.6355±0.0380 |
| | TFC | 0.5375±0.0084 | 0.5326±0.0099 | 0.5321±0.0152 | 0.5241±0.0152 | 0.5123±0.0325 | 0.5315±0.0241 |
| | TFC++ | 0.6896±0.0420 | 0.6855±0.0254 | 0.6833±0.0421 | 0.6784±0.0362 | 0.7384±0.0125 | 0.7405±0.0152 |
| | TS-TCC | 0.5105±0.0459 | 0.5137±0.0423 | 0.5228±0.0479 | 0.5105±0.0523 | 0.6006±0.0418 | 0.6000±0.0412 |
| | TS-TCC++ | 0.5613±0.0412 | 0.5614±0.0430 | 0.5540±0.0426 | 0.5442±0.0497 | 0.5714±0.0409 | 0.5804±0.0410 |
| | MixingUp | 0.5281±0.0089 | 0.5531±0.0110 | 0.5436±0.0102 | 0.5083±0.0163 | 0.5692±0.0165 | 0.5704±0.0135 |
| | MixingUp++ | 0.6418±0.0861 | 0.6696±0.0845 | 0.6309±0.0960 | 0.5954±0.0934 | 0.7119±0.0847 | 0.7109±0.0823 |
| | SLOTS (two-stage) | 0.6699±0.0452 | 0.6577±0.0432 | 0.6541±0.0422 | 0.6171±0.0428 | 0.6877±0.0512 | 0.6954±0.0510 |
| | **SLOTS (end-to-end)** | **0.7618±0.0954** | **0.7599±0.0894** | **0.7563±0.0942** | **0.7553±0.0575** | **0.8155±0.0644** | **0.8164±0.0497** |
| 40% | TS2Vec | 0.5300±0.0231 | 0.5423±0.0247 | 0.5376±0.0239 | 0.5193±0.0267 | 0.5367±0.0146 | 0.5478±0.0176 |
| | TS2Vec++ | 0.6169±0.0563 | 0.6587±0.0542 | 0.6011±0.0641 | 0.5697±0.0678 | 0.6939±0.0541 | 0.6963±0.0512 |
| | SimCLR | 0.5325±0.0224 | 0.5283±0.0236 | 0.5290±0.0245 | 0.5216±0.0278 | 0.5367±0.0214 | 0.5469±0.0228 |
| | SimCLR++ | 0.6106±0.0254 | 0.6385±0.0247 | 0.6115±0.0279 | 0.5914±0.0348 | 0.6836±0.0316 | 0.6842±0.0341 |
| | TFC | 0.5500±0.0063 | 0.5533±0.0054 | 0.5500±0.0072 | 0.5429±0.0077 | 0.5850±0.0033 | 0.5926±0.0026 |
| | TFC++ | 0.7367±0.0235 | 0.7333±0..0351 | 0.7304±0.0369 | 0.7274±0.0452 | 0.7857±0.0357 | 0.7913±0.0347 |
| | TS-TCC | 0.5525±0.0196 | 0.5288±0.0213 | 0.5308±0.0236 | 0.5183±0.0245 | 0.6032±0.0194 | 0.6016±0.0184 |
| | TS-TCC++ | 0.6021±0.0239 | 0.5991±0.0241 | 0.5883±0.0310 | 0.5768±0.0334 | 0.6320±0.0229 | 0.6332±0.0263 |
| | MixingUp | 0.5781±0.0561 | 0.5924±0.0515 | 0.5850±0.0498 | 0.5553±0.0698 | 0.6085±0.0244 | 0.6176±0.0255 |
| | MixingUp++ | 0.6479±0.0645 | 0.7101±0.0633 | 0.6653±0.0542 | 0.6211±0.0456 | 0.7730±0.0365 | 0.7707±0.0365 |
| | SLOTS (two-stage) | 0.6742±0.3254 | 0.6797±0.0412 | 0.6637±0.0523 | 0.6701±0.0489 | 0.7087±0.0421 | 0.7114±0.0432 |
| | **SLOTS (end-to-end)** | **0.8299±0.0542** | **0.8281±0.0499** | **0.8268±0.0621** | **0.8255±0.0634** | **0.8757±0.0510** | **0.8766±0.0515** |
| 60% | TS2Vec | 0.5312±0.0198 | 0.5486±0.0174 | 0.5378±0.0234 | 0.5233±0.0249 | 0.5479±0.0132 | 0.5518±0.0126 |
| | TS2Vec++ | 0.6573±0.0525 | 0.6773±0.0547 | 0.6397±0.0519 | 0.6287±0.0643 | 0.7282±0.0532 | 0.7301±0.0531 |
| | SimCLR | 0.5375±0.0361 | 0.5307±0.0348 | 0.5262±0.0396 | 0.5179±0.0348 | 0.5393±0.0362 | 0.5464±0.0352 |
| | SimCLR++ | 0.6396±0.0351 | 0.6561±0.0335 | 0.6452±0.0417 | 0.6299±0.0456 | 0.7052±0.0413 | 0.7062±0.0415 |
| | TFC | 0.5875±0.0126 | 0.5922±0.0263 | 0.5875±0.0231 | 0.5822±0.0237 | 0.6119±0.0322 | 0.6025±0.0324 |
| | TFC++ | 0.7647±0.0668 | 0.7616±0.0574 | 0.7595±0.0741 | 0.7585±0.0594 | 0.8150±0.0614 | 0.8224±0.0532 |
| | TS-TCC | 0.5625±0.0346 | 0.5663±0.0348 | 0.5611±0.0417 | 0.5413±0.0454 | 0.6091±0.0346 | 0.6027±0.0333 |
| | TS-TCC++ | 0.6026±0.0333 | 0.6058±0.0341 | 0.5900±0.0387 | 0.5792±0.0410 | 0.6404±0.0332 | 0.6413±0.0335 |
| | MixingUp | 0.6062±0.0868 | 0.6437±0.0844 | 0.6074±0.0725 | 0.6011±0.0769 | 0.6159±0.0631 | 0.6199±0.0612 |
| | MixingUp++ | 0.6960±0.0621 | 0.7304±0.0555 | 0.6731±0.0635 | 0.6392±0.0789 | 0.7837±0.0541 | 0.7845±0.0569 |
| | SLOTS (two-stage) | 0.7054±0.0535 | 0.7038±0.0478 | 0.7275±0.0621 | 0.6975±0.0598 | 0.7610±0.0459 | 0.7564±0.0425 |
| | **SLOTS (end-to-end)** | **0.8728±0.0625** | **0.8719±0.0654** | **0.8703±0.0968** | **0.8696±0.0842** | **0.9101±0.0682** | **0.9112±0.0574** |
| 80% | TS2Vec | 0.5475±0.0213 | 0.5368±0.0241 | 0.5355±0.0219 | 0.5341±0.0238 | 0.5491±0.0196 | 0.5549±0.0194 |
| | TS2Vec++ | 0.6886±0.0436 | 0.7112±0.0479 | 0.6672±0.0521 | 0.6587±0.0642 | 0.7704±0.0534 | 0.7691±0.0537 |
| | SimCLR | 0.5400±0.0169 | 0.5677±0.0198 | 0.5640±0.0246 | 0.5383±0.0234 | 0.5365±0.0167 | 0.5457±0.0165 |
| | SimCLR++ | 0.6958±0.0517 | 0.6994±0.0514 | 0.6847±0.0496 | 0.6823±0.0463 | 0.7641±0.0521 | 0.7575±0.0530 |
| | TFC | 0.6025±0.0168 | 0.6048±0.0167 | 0.5918±0.0215 | 0.5926±0.0266 | 0.6212±0.0214 | 0.6470±0.0215 |
| | TFC++ | 0.8000±0.0354 | 0.7967±0.0471 | 0.7953±0.0532 | 0.7939±0.0613 | 0.8403±0.0536 | 0.8459±0.0475 |
| | TS-TCC | 0.5675±0.0310 | 0.5701±0.0296 | 0.5697±0.0309 | 0.5529±0.0342 | 0.6115±0.0296 | 0.6092±0.0284 |
| | TS-TCC++ | 0.6028±0.0298 | 0.6077±0.0310 | 0.5933±0.0301 | 0.5889±0.0364 | 0.6492±0.0263 | 0.6420±0.0254 |
| | MixingUp | 0.6500±0.0551 | 0.6540±0.0351 | 0.6529±0.0532 | 0.6498±0.0647 | 0.6679±0.0345 | 0.6547±0.0367 |
| | MixingUp++ | 0.6973±0.0542 | 0.7443±0.0596 | 0.6804±0.0496 | 0.6522±0.0515 | 0.8091±0.0362 | 0.8090±0.0337 |
| | SLOTS (two-stage) | 0.7312±0.0625 | 0.7460±0.0654 | 0.7520±0.0741 | 0.7615±0.0632 | 0.7812±0.0678 | 0.8210±0.0593 |
| | **SLOTS (end-to-end)** | **0.9037±0.0774** | **0.9028±0.0710** | **0.9028±0.0821** | **0.9020±0.0736** | **0.9282±0.0630** | **0.9294±0.0622** |
| 100% | TS2Vec | 0.5499±0.0225 | 0.5373±0.0172 | 0.5376±0.0200 | 0.5478±0.0235 | 0.5597±0.0046 | 0.5788±0.0050 |
| | TS2Vec++ | 0.6982±0.0720 | 0.7160±0.0716 | 0.6684±0.0718 | 0.6629±0.0726 | 0.7726±0.0912 | 0.7810±0.0866 |
| | SimCLR | 0.5687±0.0212 | 0.5728±0.0235 | 0.5682±0.0194 | 0.5460±0.0198 | 0.5694±0.0157 | 0.5524±0.0096 |
| | SimCLR++ | 0.7087±0.0412 | 0.7028±0.0455 | 0.6882±0.0494 | 0.6860±0.0497 | 0.7694±0.0453 | 0.7594±0.0481 |
| | TFC | 0.6225±0.0025 | 0.6181±0.0065 | 0.6475±0.0043 | 0.6037±0.0047 | 0.6538±0.0191 | 0.6513±0.0223 |
| | TFC++ | 0.8167±0.0914 | 0.8074±0.0902 | 0.8125±0.0958 | 0.8118±0.0959 | 0.8504±0.0772 | 0.8526±0.0750 |
| | TS-TCC | 0.5725±0.0150 | 0.5706±0.0171 | 0.5713±0.0119 | 0.5558±0.0065 | 0.6185±0.0014 | 0.6161±0.0056 |
| | TS-TCC++ | 0.6184±0.0292 | 0.6209±0.0371 | 0.5996±0.0239 | 0.5897±0.0226 | 0.6606±0.0474 | 0.6610±0.0439 |
| | MixingUp | 0.6562±0.0062 | 0.6610±0.0082 | 0.6504±0.0033 | 0.6505±0.0154 | 0.6717±0.0364 | 0.6852±0.0271 |
| | MixingUp++ | 0.7499± 0.1017 | 0.7576±0.1037 | 0.7430± 0.1036 | 0.7372±0.1112 | 0.7050±0.0924 | 0.7048±0.0926 |
| | SLOTS (two-stage) | 0.7525±0.0500 | 0.7594±0.0585 | 0.7401±0.0398 | 0.7664±0.0415 | 0.8197±0.0067 | 0.8388±0.0087 |
| | **SLOTS (end-to-end)** | **0.9342±0.1161** | **0.9329±0.1171** | **0.9335±0.1184** | **0.9327± 0.1187** | **0.9456±0.0966** | **0.9469±0.0952** |

Table 2: Performance comparison on SEED, P19, EPILEPSY, and HAR, under the label ratio of 10%. To save space, we show the extended table with errorbars in Appendix C.

| Models | SEED | | | | | | P19 | | | | | |
|---|---|---|---|---|---|---|---|---|---|---|---|---|
| | Accuracy | Precision | Recall | F1 score | AUROC | AUPRC | Accuracy | Precision | Recall | F1 score | AUROC | AUPRC |
| TS2Vec | 0.5750 | 0.6048 | 0.5732 | 0.5113 | 0.6493 | 0.6218 | 0.6580 | 0.6467 | 0.5520 | 0.5669 | 0.5171 | 0.5860 |
| TS2Vec++ | 0.6252 | 0.6191 | 0.6084 | 0.5599 | 0.6607 | 0.6962 | 0.6589 | 0.6794 | 0.6500 | 0.6894 | 0.6527 | 0.6978 |
| TS-TCC | 0.5893 | 0.5627 | 0.5772 | 0.5723 | 0.7137 | 0.6941 | 0.9216 | 0.5968 | 0.6597 | 0.6156 | 0.6941 | 0.6667 |
| TS-TCC++ | 0.6172 | 0.5935 | 0.5776 | 0.5977 | 0.8004 | **0.7637** | **0.9654** | 0.7338 | 0.6776 | 0.7076 | 0.7011 | 0.7065 |
| SimCLR | 0.5050 | 0.5115 | 0.5313 | 0.5155 | 0.6963 | 0.6982 | 0.9130 | 0.5877 | 0.5600 | 0.5338 | 0.5995 | 0.5956 |
| SimCLR++ | 0.5736 | 0.5344 | 0.5505 | 0.5374 | 0.6802 | 0.6361 | 0.9534 | 0.6924 | 0.6124 | 0.6321 | 0.7077 | 0.6554 |
| MixingUp | 0.5166 | 0.5356 | 0.5638 | 0.5141 | 0.6605 | 0.6580 | 0.9136 | 0.6744 | 0.6481 | 0.6477 | 0.6802 | 0.6361 |
| MixingUp++ | 0.5569 | 0.5701 | 0.5638 | 0.5265 | 0.7494 | 0.6603 | 0.9253 | 0.6777 | 0.6500 | 0.6489 | 0.7685 | 0.6932 |
| TFC | 0.5500 | 0.5356 | 0.5487 | 0.5290 | 0.7499 | 0.6423 | 0.9049 | 0.6207 | 0.6340 | 0.6267 | 0.6399 | 0.7092 |
| TFC++ | 0.6087 | 0.6121 | 0.6071 | 0.5990 | 0.7847 | 0.6685 | 0.9361 | 0.7500 | 0.6761 | 0.6954 | **0.7721** | 0.7287 |
| SLOTS (two-stage) | 0.6468 | 0.6573 | 0.7316 | 0.6169 | 0.7575 | 0.7572 | 0.9254 | 0.7138 | 0.6776 | 0.7011 | 0.7065 | 0.7065 |
| SLOTS (end-to-end) | **0.7181** | **0.7286** | **0.7869** | **0.6510** | **0.8034** | 0.7592 | 0.9596 | **0.7580** | **0.7416** | **0.7788** | 0.7703 | **0.8270** |

| Models | EPILEPSY | | | | | | HAR | | | | | |
|---|---|---|---|---|---|---|---|---|---|---|---|---|
| | Accuracy | Precision | Recall | F1 score | AUROC | AUPRC | Accuracy | Precision | Recall | F1 score | AUROC | AUPRC |
| TS2Vec | 0.5125 | 0.5604 | 0.5118 | 0.5287 | 0.6519 | 0.6233 | 0.5738 | 0.5555 | 0.5768 | 0.5565 | 0.7623 | 0.7413 |
| TS2Vec++ | 0.6022 | 0.6011 | 0.5500 | 0.5445 | 0.6537 | 0.6498 | 0.6743 | 0.6197 | 0.5838 | 0.5701 | 0.8488 | 0.8114 |
| TS-TCC | 0.6175 | 0.6338 | 0.5000 | 0.5403 | 0.6029 | 0.6726 | 0.6596 | 0.6154 | 0.6158 | 0.5459 | 0.7511 | 0.7414 |
| TS-TCC++ | 0.6206 | 0.6401 | 0.5459 | 0.5445 | 0.6851 | 0.7150 | 0.6787 | 0.6473 | 0.6316 | 0.6169 | 0.7575 | 0.7572 |
| SimCLR | 0.6375 | 0.5938 | 0.5200 | 0.5100 | 0.6293 | 0.6776 | 0.6168 | 0.6390 | 0.5704 | 0.5744 | 0.7452 | 0.7098 |
| SimCLR++ | 0.6805 | 0.6591 | 0.5084 | 0.5599 | 0.6607 | 0.6962 | 0.6217 | 0.6435 | 0.6205 | 0.5977 | 0.9004 | 0.7637 |
| MixingUp | 0.6175 | 0.6115 | 0.5313 | 0.5485 | 0.6596 | 0.6782 | 0.6277 | 0.6393 | 0.6138 | 0.6055 | 0.7705 | 0.7481 |
| MixingUp++ | 0.6617 | 0.6356 | 0.5804 | 0.5734 | 0.6605 | 0.6980 | 0.6699 | 0.6577 | 0.6541 | 0.6172 | 0.8878 | 0.8254 |
| TFC | 0.6550 | 0.6312 | 0.5250 | 0.5226 | 0.5390 | 0.6653 | 0.6502 | 0.6102 | 0.6083 | 0.6095 | 0.7315 | 0.7371 |
| TFC++ | 0.6693 | 0.6627 | 0.5472 | 0.5472 | 0.7014 | 0.6834 | 0.7263 | 0.6479 | 0.6634 | 0.6512 | 0.8519 | 0.8064 |
| SLOTS (two-stage) | 0.6652 | 0.6591 | 0.5684 | 0.5599 | 0.6607 | 0.6962 | 0.6654 | 0.6538 | 0.6276 | 0.6076 | 0.8011 | 0.7565 |
| SLOTS (end-to-end) | **0.6955** | **0.6952** | **0.5999** | **0.5819** | **0.7105** | **0.7600** | **0.7312** | **0.6661** | **0.6720** | **0.6615** | **0.9013** | **0.8811** |

Table 3: Ablation study (DEAP; 10% label ratio). 'W/o $\mathcal{L}_u$' means removing unsupervised contrastive loss, i.e., encoders and classifiers are updated with supervised loss and cross-entropy loss. 'W/o $\mathcal{L}_s$' refers to removing supervised contrastive loss, i.e., encoders and classifiers are updated with unsupervised loss and classification loss. 'W/ $\mathcal{L}_s$' refers to introducing supervised contrastive loss in SLOTS's fine-tuning stage.

| Models | Methods | Accuracy | Precision | Recall | F1 score | AUROC | AUPRC |
|---|---|---|---|---|---|---|---|
| SLOTS (two-stage) | W/ $\mathcal{L}_s$ in fine-tuning | 0.6375±0.0100 | 0.6433±0.0255 | 0.6359±0.0130 | 0.6308±0.0159 | 0.6897±0.0061 | 0.6865±0.0067 |
| | Full model | 0.6276±0.0541 | 0.6393±0.0495 | 0.6138±0.0547 | 0.6055±0.0563 | 0.6704±0.0471 | 0.6480±0.0396 |
| SLOTS (end-to-end) | W/o $\mathcal{L}_u$ | 0.6590±0.0724 | 0.6548±0.0757 | 0.6490±0.0736 | 0.6472±0.0748 | 0.7042±0.0961 | 0.7049±0.0927 |
| | W/o $\mathcal{L}_s$ | 0.6103±0.0520 | 0.6072±0.0583 | 0.5953±0.0497 | 0.5905±0.0495 | 0.6438±0.0771 | 0.6469±0.0714 |
| | Full model | **0.6636±0.0763** | **0.6598±0.0791** | **0.6571± 0.0773** | **0.6556±0.0781** | **0.7115±0.0995** | **0.7118±0.0953** |

F1 score. (1) The results demonstrate that transitioning a model from a two-stage to its end-to-end version (without any other changes) leads to significant performance improvement. For instance, at an 80% label ratio, TFC++ achieves the highest improvement over TFC, with a 20.13% increase. (2) Furthermore, our end-to-end SLOTS model consistently outmatchs its two-stage counterpart, and the performance margin increases as the labeling ratio increases. Specifically, with 100% labeled data, SLOTS (end-to-end) achieved a significant improvement of 16.63% over SLOTS (two-stage).

## 4.2 COMPARING MODEL: SLOTS VERSUS BASELINES

Our end-to-end SLOTS model consistently outperforms all ten baselines. With a full availably of data labels, SLOTS (end-to-end) achieves the highest performance of 93.27%, surpassing the best baseline, TFC++ (81.18%), by a significant margin of 12.09% (Table 1). Additionally, our model outperforms all baselines across multiple datasets, including SEED, P19, EPILEPSY, and HAR datasets (Table 2). Overall, our SLOTS (end-to-end) model achieves victory in 21 out of 24 tests (6 metrics in 4 datasets) and is the second-best performer in 3 other tests.

**Ablation study** To demonstrate the importance of each loss component, we examine two scenarios: 1) removing unsupervised losses; 2) removing supervised losses. Both scenarios result in degraded performance (DEAP; 10% labeling ratio; Table 3)). Additionally, we incorporate a supervised contrastive loss into the two-stage version of SLOTS, which leads to a slight increase (1% in accuracy) but still remains 2.6% lower than the end-to-end framework.

Table 4: Performance comparison (DEAP; 40% label ratio) across trial-dependent, leave-trials-out, and leave-subjects-out settings. We present the comparison with the strongest baseline on DEAP (MixingUp) to save space.

| Patterns | Models | Accuracy | Precision | Recall | F1 Score | AUROC | AUPRC |
|---|---|---|---|---|---|---|---|
| **Trial-dependent** | MixingUp | 0.6563±0.0012 | 0.6692±0.0065 | 0.6499±0.0111 | 0.6340±0.0047 | 0.7400±0.0058 | 0.7262±0.0052 |
| | MixingUp++ | 0.8670±0.0738 | 0.8827±0.0638 | 0.8676±0.0698 | 0.8612±0.0777 | 0.9548±0.0371 | 0.9537±0.0371 |
| | SLOTS (end-to-end) | **0.9418±0.0279** | **0.9410±0.0280** | **0.9410±0.0287** | **0.9404±0.0286** | **0.9818±0.0134** | **0.9817±0.0132** |
| **Leave-trials-out** | MixingUp | 0.5781±0.0561 | 0.5924±0.0515 | 0.5850±0.0498 | 0.5553±0.0698 | 0.6085±0.0244 | 0.6176±0.0255 |
| | MixingUp++ | 0.6479±0.0645 | 0.7101±0.0633 | 0.6653±0.0542 | 0.6211±0.0655 | 0.7730±0.0365 | 0.7707±0.0365 |
| | SLOTS (end-to-end) | **0.8299±0.0542** | **0.8281±0.0499** | **0.8268±0.0621** | **0.8255±0.0634** | **0.8757±0.0510** | **0.8766±0.0515** |
| **Leave-subjects-out** | MixingUp | 0.5062±0.0038 | 0.5202±0.0017 | 0.5186±0.0107 | 0.5104±0.0085 | 0.5878±0.0118 | 0.5891±0.0086 |
| | MixingUp++ | 0.6085±0.0136 | 0.6640±0.0182 | 0.6278±0.0088 | 0.5788±0.0121 | 0.7010±0.0106 | 0.7006±0.0099 |
| | SLOTS (end-to-end) | **0.6765±0.1204** | **0.6864±0.1175** | **0.6818±0.1156** | **0.6708±0.1200** | **0.7330±0.1342** | **0.7365±0.1280** |

### 4.3 ROBUSTNESS ACROSS TRIALS AND SUBJECTS

Taking emotion recognition (DEAP; 40% labels) as an example, we investigate the robustness of SLOTS in three settings (Appendix D): trial-dependent, leave-trials-out, and leave-subjects-out. We compare SLOTS with MixingUp which is the strongest baseline on DEAP (Table 1). In Table 4, as expected, the trial-dependent pattern outstrips leave-trials-out, which in turn outperforms the leave-subjects-out pattern. This could be due to inter-trial and inter-subject variations Shen et al. (2022). In the trial-dependent pattern, SLOTS (end-to-end) achieves the highest F1 score of 94.04%, surpassing MixingUp by 30.64% and MixingUp++ by 7.92%. SLOTS (end-to-end) consistently performs best across all three patterns and six metrics, demonstrating the effectiveness of SLOTS.

## 5 DISCUSSION AND FUTURE WORKS

SLOTS demonstrated promising results across various datasets, outperforming ten competitive baselines. Nevertheless, there are several aspects that warrant further investigation and improvements.

Firstly, the methodology employed in our approach prioritizes enhanced performance at the expense of transferability. Unlike two-stage models that facilitate pre-training on a distinct dataset followed by fine-tuning on a smaller dataset, our end-to-end model necessitates that both pre-training and fine-tuning datasets be derived from the same source. This constraint hampers the model's capacity to transfer knowledge across disparate tasks and datasets. A potential remedy to this issue may entail incorporating domain adaptation techniques, as delineated by Wilson et al. (2020) Wilson et al. (2020), to address both model performance and knowledge transferability concurrently.

Secondly, the existing model offers opportunities for extension to a wider array of downstream tasks. In principle, our end-to-end framework can be tailored to accommodate tasks such as clustering, forecasting, regression, anomaly detection, and other time series data tasks, granted that an appropriate task-specific loss function is incorporated.

Lastly, the data augmentation techniques employed in our model warrant further refinement Liu et al. (2023). The development of more sophisticated and task-specific augmentation methods could yield superior representations, consequently enhancing the model's performance.

## 6 CONCLUSION

This paper presented SLOTS, an end-to-end model for semi-supervised time series classification, which combines unsupervised contrastive loss, supervised contrastive loss, and classification loss in a unified framework. The hybrid loss function effectively leverages both labeled and unlabeled data to learn discriminative features, while the end-to-end training process allows the model to adaptively balance the contributions of these losses to achieve optimal performance. Extensive experiments demonstrated that SLOTS surpasses state-of-the-art methods and their end-to-end adaptations, showcasing its effectiveness and versatility in handling time series classification tasks.

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

## APPENDIX A    DATASET DETAILS

We use five diverse time series datasets to evaluate our model. Following is a detailed description of the datasets.

DEAP dataset Koelstra et al. (2012). The publicly available database DEAP consists of a multimodal dataset for the analysis of human emotional states. A total of 32 EEG channels and eight peripheral physiological signals of 32 subjects (aged between 19 and 37) were recorded whilst watching music videos with a sampling frequency of 128 Hz. Then the recorded signals are passed from a bandpass filter to remove noise and artifacts like eye blinks. The 40 one-minute-long videos were carefully selected to elicit different emotional states according to the dimensional, valence-arousal. The valence-arousal emotion model, first proposed by Russell Russell (1980), places each emotional state on a two-dimensional scale. The first dimension represents valence, which ranged from negative to positive, and the second dimension is arousal, which ranged from calm to exciting. In DEAP, each video clip is rated from 1 to 9 for arousal and valence by each subject after the viewing, and the discrete rating value can be used as a classification label in emotion recognition Frydenlund & Rudzicz (2015); Pham et al. (2015); Lan et al. (2016); Chao & Dong (2021); Bagherzadeh et al. (2022).

SEED dataset Zheng et al. (2019). The SJTU Emotion EEG Dataset (SEED) is a free and publicly available EEG dataset for emotional analysis provided by Shanghai Jiao Tong University in 2015. SEED dataset provided EEG recordings from 15 subjects (8 females and 7 males, 23.27 ± 2.37 years) using a 62-channel system. In total, 45 trials (the same 15 movie clips repeated on 3 days) were presented to the subjects. Each trial lasted for approximately 4 minutes. Each participant contributed to the experiment thrice at an interval of one week or longer. For data preprocessing, these recordings were downsampled from 1000 to 200 Hz and filtered to 0.5–70 Hz. In addition, the labels of the trials were obtained from the self-assessments, and the elicitations involved three emotional categories: positive, neutral, and negative.

P19 dataset Reyna et al. (2020). P19 (PhysioNet Sepsis Early Prediction Challenge 2019) dataset contains 38,803 patients and each patient is monitored by 34 irregularly sampled sensors including 8 vital signs and 26 laboratory values. The original dataset has 40,336 patients, we remove the samples with too short or too long time series, remaining 38,803 patients (the longest time series of the patient has more than one and less than 60 observations). Each patient is associated with a static vector indicating attributes: age, gender, time between hospital admission and ICU admission, ICU type, and ICU length of stay (days). Each patient has a binary label representing occurrence of sepsis within the next 6 hours. The dataset is highly imbalanced with only ~4% positive samples. Raw data of P19 can be found at https://physionet.org/content/challenge-2019/1.0.0/.

EPILEPSY dataset Andrzejak et al. (2001). The dataset contains single-channel EEG measurements from 500 subjects. For every subject, the brain activity was recorded for 23.6 seconds. The dataset was then divided and shuffled (to mitigate sample-subject association) into 11,500 samples of 1 second each, sampled at 178 Hz. The raw dataset features five classification labels corresponding to different states of subjects or measurement locations — eyes open, eyes closed, EEG measured in the healthy brain region, EEG measured in the tumor region, and whether the subject has a seizure episode. To emphasize the distinction between positive and negative samples in terms of epilepsy, We merge the first four classes into one, and each time series sample has a binary label describing if the associated subject is experiencing a seizure or not. There are 11,500 EEG samples in total. The raw dataset (https://repositori.upf.edu/handle/10230/42894) is distributed under the Creative Commons License (CC-BY) 4.0.

HAR Anguita et al. (2013). This dataset contains recordings of 30 health volunteers performing daily activities, including walking, walking upstairs, walking downstairs, sitting, standing, and lying. Prediction labels are the six activities. The wearable sensors on a smartphone measure triaxial linear acceleration and triaxial angular velocity at 50 Hz. After preprocessing and isolating gravitational acceleration from body acceleration, there are nine channels (i.e., 3-axis accelerometer, 3-axis gyroscope, and 3-axis magnetometer) in total. The raw dataset (https://archive.ics.uci.edu/ml/datasets/Human+Activity+Recognition+Using+Smartphones) is distributed as-is. Any commercial use is not allowed.

## APPENDIX B   EXPERIMENTAL DETAILS

We implement the baselines follow the corresponding papers including TS2vec Yue et al. (2022), Mixing-up Wickstrøm et al. (2022), TS-TCC Eldele et al. (2021), SimCLR Chen et al. (2020), and TFC Zhang et al. (2022a). Unless noted below, we use default settings for hyper-parameters as reported in the original works. All two-stage and end-to-end frameworks with baselines are done with an NVIDIA GeForce RTX 3090 GPU with 256 GB of allocated memory.

SLOTS (our model) allows for the simultaneous training of the encoder and classifier, unifying model optimization, reducing intermediate computational resources, and learning task-specific features. Our encoder $f$ is similar to ECNN-C Li et al. (2022a). Specifically, the novel block structure is designed to comprise four different convolutional layers. First, we adopt two consecutive kernels $3 \times 1$ and $1 \times 3$ to replace the convolution kernel of shape of $3 \times 3$ by taking a similar operation to MobileNet model. Second, we use spatial convolution with a depth multiplier of 2, thus doubling the number of channels in the first deep convolution layer. The purpose of this approach is to enable the model to have sufficient channels, allowing them to utilize a reduction factor in bottleneck structures Freeman et al. (2018). Third, separable pooling is used to replace the max-pooling due to the aliasing in the step convolution. We adopt a $2 \times 1$ pooling kernel after the first spatial convolutional layer. The classifier contains one fully-connected layer. We employ a specific data augmentation technique for the proposed algorithm. This involves masking the data using a binary mask, where each element is independently sampled from a Bernoulli distribution with a probability of p=0.5. We use a batch size of 100 and a training epoch of 30.

TS2Vec Yue et al. (2022) introduces the notion of contextual consistency and uses a hierarchical loss function to capture long-range structure in time series. TS2vec is a powerful representative learning method and has a specially-designed architecture. The encoder network consists of three components. First, the input time series is augmented by selecting overlapping subseries. They are projected into a higher dimensional latent space. Then, latent vectors for input time series are masked at randomly chosen positions. Finally, a dilated CNN with residual blocks produces the contextual representations. To compute the loss, the representations are gradually pooled along the time dimension and at each step a loss function based on contextual consistency is applied. For the baseline experiment, we found that the original 10 layers of ResNet blocks are redundant, and we reduce residual blocks in the encoder from 10-layer to 3-layer without compromising model performances.

Mixing-up Wickstrøm et al. (2022) proposes new mixing-up augmentation and pretext tasks that aim to correctly predict the mixing proportion of two time series samples. In Mixing-up, the augmentation is chosen as the convex combination of two randomly drawn time series from the dataset, where the mixing parameter is random drawn from a beta distribution. The contrastive loss is then computed between the two inputs and the augmented time series. The loss is a minor modification of NT-Xent loss and is designed to encourage the correct prediction of the amount of mixing. We use the same beta distribution as reported in the original Mixing-up model.

TS-TCC Eldele et al. (2021) leverages contextual information with a transformer-based autoregressive model and ensures transferability by using both strong and weak augmentations. TS-TCC proposed a challenging pretext task. An input time series sample is first augmented by adding noise, scaling, and permuting time series. The views are then passed through an encoder consisting of three convolutional layers before being processed by the temporal contrasting module. During temporal contrasting, for each view, a transformer architecture is used to learn a contextual representation. The learned representation is then used to predict latent observation of the other augmented view at a future time. The contextual representations are then projected and maximized similarity using NT-Xent loss. For this baseline, we mostly adopted the hyper-parameters presented in the original paper.

SimCLR Chen et al. (2020) is a state-of-the-art model in self-supervised representation learning of images. It utilizes deep learning architectures to generate augmentation-based embeddings and optimize the model parameters by minimizing NT-Xent loss in the embedding space. Although initially proposed for image data, it is readily adapted to time series, as shown in Eldele et al. (2021). An input time series sample is first augmented into two related views. Then a base encoder extracts representation vectors. ResNet is used as an encoder backbone for simplicity. Then the projection head transforms representations into a latent space where the NT-Xent loss is applied. For time series, SimCLR investigated two augmentations, including jitter-and-scale, and permutation-and-jitter. All the unmentioned hyper-parameters are kept the same as the original model.

TFC Zhang et al. (2022a) introduces a strategy for self-supervised pre-training in time series by modeling Time-Frequency Consistency. TFC specifies that time-based and frequency-based representations, learned from the same time series sample, should be closer to each other in the time-frequency space than representations of different time series samples. Specifically, contrastive learning in time-space is adapted to generate a time-based representation. In parallel, TFC proposes a set of novel augmentations based on the characteristic of the frequency spectrum and produces a frequency-based embedding through contrastive instance discrimination. All the unmentioned hyper-parameters are kept the same as the original model.

## APPENDIX C  EXTENDED EXPERIMENTAL RESULTS

Table 5 is the performance with errorbars comparison on SEED, P19, EPILEPSY, and HAR, under the label ratio of 10%.

## APPENDIX D  ADDITIONAL INFORMATION ON ROBUSTNESS VALIDATION

We investigate the robustness of SLOTS in three settings on DEAP dataset: trial-dependent, leave-trials-out, and leave-subjects-out Liang et al. (2021); Kumari et al. (2022); Kim & Jo (2020); Mert & Akan (2018). The details of these three settings are explained as follows.

1) Trial-dependent

In the DEAP dataset, we permute the trials of 32 subjects and randomly selected some trials as the training set and the remaining trials as the testing set.

2) Leave-trials-out

In the DEAP dataset, each subject has 40 trials. We take out some of their trials for each of the 32 subjects to form the training set, and the remaining trials to form the testing set. For example, the first 36 trials are used as the training set and the last 4 trials as the testing set. The total number of trials in the training set is $36 \times 32 = 1152$; the total number of trials in the testing set is $4 \times 32 = 128$.

3) Leave-subjects-out

In the DEAP dataset, there are a total of 32 subjects. We randomly select a part of the subjects as the training set and a part of the subjects as the testing set. For example, 30 subjects are randomly selected as the training set and the remaining 2 subjects as the testing set. The total number of trials in the training set is $30 \times 40 = 1200$; the number of trials in the test set is $2 \times 40 = 80$.

## APPENDIX E  TRANSFERABILITY OF SLOTS

We note the trade-off between adaptability and model performance. While two-stage models may be more effective in transfer learning or situations where pre-training and fine-tuning involve different datasets, our SLOTS model is a superior choice for achieving optimal performance on a specific dataset (Section 5).

In the two-stage framework, we use the DEAP dataset during pre-training and the SEED dataset during fine-tuning. We report metrics including accuracy, precision (macro-averaged), recall, F1-score, AUROC (one-versus-rest), and AUPRC. The evaluation results are shown in Table 6.

Table 5: Performance with errorbars comparison on SEED, P19, EPILEPSY, and HAR, under the label ratio of 10%.

| Models | SEED | | | | | |
|---|---|---|---|---|---|---|
| | Accuracy | Precision | Recall | F1 score | AUROC | AUPRC |
| TS2Vec | 0.5750 ±0.0561 | 0.6048 ±0.0751 | 0.5732 ±0.0836 | 0.5113 ±0.1023 | 0.6493 ±0.0651 | 0.6218 ±0.0712 |
| TS2Vec++ | 0.6252 ±0.0695 | 0.6191 ±0.0752 | 0.6084 ±0.0658 | 0.5599 ±0.0832 | 0.6607 ±0.0694 | 0.6962 ±0.0769 |
| TS-TCC | 0.5893 ±0.0852 | 0.5627 ±0.0935 | 0.5772 ±0.0876 | 0.5723 ±0.0987 | 0.7137 ±0.0876 | 0.6941 ±0.0635 |
| TS-TCC++ | 0.6172 ±0.0845 | 0.5935 ±0.0657 | 0.5776 ±0.0589 | 0.5977 ±0.0896 | 0.8004 ±0.0674 | **0.7637 ±0.0863** |
| SimCLR | 0.5050 ±0.0651 | 0.5115 ±0.0769 | 0.5313 ±0.0984 | 0.5155 ±0.0863 | 0.6963 ±0.0785 | 0.6982 ±0.0895 |
| SimCLR++ | 0.5736 ±0.0782 | 0.5344 ±0.0972 | 0.5505 ±0.0871 | 0.5374 ±0.0777 | 0.6802 ±0.0698 | 0.6361 ±0.0974 |
| MixUp | 0.5166 ±0.0642 | 0.5356 ±0.0759 | 0.5638 ±0.0968 | 0.5141±0.0874 | 0.6605 ±0.0778 | 0.6580 ±0.0965 |
| MixUp++ | 0.5569 ±0.0869 | 0.5701 ±0.0759 | 0.5638 ±0.0963 | 0.5265 ±0.0758 | 0.7494 ±0.0851 | 0.6603 ±0.0685 |
| TFC | 0.5500 ±0.0542 | 0.5356 ±0.0693 | 0.5487 ±0.0579 | 0.5290 ±0.0698 | 0.7499 ±0.0782 | 0.6423 ±0.0869 |
| TFC++ | 0.6087 ±0.0368 | 0.6071 ±0.0597 | 0.6071 ±0.0698 | 0.5990 ±0.0475 | 0.7847 ±0.0369 | 0.6685 ±0.0475 |
| SLOTS(two-stage) | 0.6468 ±0.0682 | 0.6573 ±0.0864 | 0.7316 ±0.0987 | 0.6169 ±0.0996 | 0.7575 ±0.0874 | 0.7572 ±0.0872 |
| SLOTS(end-to-end) | **0.7181 ±0.1023** | **0.7286 ±0.0964** | **0.7869 ±0.0867** | **0.6510 ±0.1136** | **0.8034 ±0.0987** | 0.7592 ±0.0869 |

| Models | P19 | | | | | |
|---|---|---|---|---|---|---|
| | Accuracy | Precision | Recall | F1 score | AUROC | AUPRC |
| TS2Vec | 0.6580 ±0.0562 | 0.6467 ±0.0634 | 0.5520 ±0.0675 | 0.5669 ±0.0741 | 0.5171 ±0.0683 | 0.5860 ±0.0652 |
| TS2Vec++ | 0.6589 ±0.0358 | 0.6794 ±0.0210 | 0.6500 ±0.0235 | 0.6894 ±0.0355 | 0.6527 ±0.0196 | 0.6978 ±0.0210 |
| TS-TCC | 0.9216 ±0.0325 | 0.5968 ±0.0413 | 0.6597 ±0.0321 | 0.6156 ±0.0369 | 0.6941 ±0.0299 | 0.6667 ±0.0345 |
| TS-TCC++ | **0.9654 ±0.0681** | 0.7338 ±0.0593 | 0.6776 ±0.0581 | 0.7076 ±0.0698 | 0.7011 ±0.0782 | 0.7065 ±0.0635 |
| SimCLR | 0.9130 ±0.0594 | 0.5877 ±0.0425 | 0.5600 ±0.0501 | 0.5338 ±0.0632 | 0.5995 ±0.0596 | 0.5956 ±0.0720 |
| SimCLR++ | 0.9534 ±0.0368 | 0.6924 ±0.0244 | 0.6124 ±0.0543 | 0.6321 ±0.0576 | 0.7077 ±0.0147 | 0.6554 ±0.0247 |
| MixUp | 0.9136 ±0.0369 | 0.6744 ±0.0475 | 0.6481 ±0.0355 | 0.6477 ±0.0593 | 0.6802 ±0.0682 | 0.6361 ±0.0651 |
| MixUp++ | 0.9253 ±0.0741 | 0.6777 ±0.0655 | 0.6500 ±0.0357 | 0.6489 ±0.0698 | 0.7685 ±0.0241 | 0.6932 ±0.0369 |
| TFC | 0.9049 ±0.0357 | 0.6207 ±0.0366 | 0.6340 ±0.0445 | 0.6267 ±0.0369 | 0.6399 ±0.0485 | 0.7092 ±0.0436 |
| TFC++ | 0.9361 ±0.0563 | 0.7500 ±0.0684 | 0.6761 ±0.0536 | 0.6954 ±0.0458 | **0.7721 ±0.0412** | 0.7287 ±0.0369 |
| SLOTS(two-stage) | 0.9254 ±0.0842 | 0.7138 ±0.0933 | 0.6776 ±0.0789 | 0.7076 ±0.1023 | 0.7011 ±0.0698 | 0.7065 ±0.0563 |
| SLOTS(end-to-end) | 0.9596 ±0.1036 | **0.7580 ±0.1025** | **0.7416 ±0.1124** | **0.7788 ±0.1253** | 0.7703 ±0.0865 | **0.8270 ±0.0869** |

| Models | EPILEPSY | | | | | |
|---|---|---|---|---|---|---|
| | Accuracy | Precision | Recall | F1 score | AUROC | AUPRC |
| TS2Vec | 0.5125 ±0.0569 | 0.5604 ±0.0647 | 0.5118 ±0.0541 | 0.5287 ±0.0698 | 0.6519 ±0.0754 | 0.6233 ±0.0694 |
| TS2Vec++ | 0.6022 ±0.0658 | 0.6011 ±0.0794 | 0.5500 ±0.0635 | 0.5445 ±0.0774 | 0.6537 ±0.0896 | 0.6498 ±0.0894 |
| TS-TCC | 0.6175 ±0.0313 | 0.6338 ±0.0779 | 0.5000 ±0.0312 | 0.5403 ±0.0510 | 0.6029 ±0.0122 | 0.6726 ±0.0097 |
| TS-TCC++ | 0.6206 ±0.0818 | 0.6401 ±0.0938 | 0.5459 ±0.0936 | 0.5445 ±0.0928 | 0.6851 ±0.0678 | 0.7150 ±0.0715 |
| SimCLR | 0.6375 ±0.0751 | 0.5938 ±0.0698 | 0.5200 ±0.0876 | 0.5100 ±0.0964 | 0.6293 ±0.0854 | 0.6776 ±0.0769 |
| SimCLR++ | 0.6805 ±0.0984 | 0.6591 ±0.0842 | 0.5084 ±0.0687 | 0.5599 ±0.0964 | 0.6607 ±0.0578 | 0.6962 ±0.0635 |
| MixUp | 0.6175 ±0.0778 | 0.6115 ±0.0863 | 0.5313 ±0.0684 | 0.5485 ±0.0851 | 0.6596 ±0.0657 | 0.6782 ±0.0423 |
| MixUp++ | 0.6617 ±0.0412 | 0.6356 ±0.0324 | 0.5804 ±0.0256 | 0.5734 ±0.0421 | 0.6605 ±0.0332 | 0.6980 ±0.0236 |
| TFC | 0.6550 ±0.0369 | 0.6312 ±0.0475 | 0.5250 ±0.0452 | 0.5226 ±0.0637 | 0.5390 ±0.0635 | 0.6653 ±0.0569 |
| TFC++ | 0.6693 ±0.0786 | 0.6627 ±0.0684 | 0.5472 ±0.0756 | 0.5472 ±0.0692 | 0.7014 ±0.0462 | 0.6834 ±0.0553 |
| SLOTS(two-stage) | 0.6652 ±0.0896 | 0.6591 ±0.0974 | 0.5684 ±0.0963 | 0.5599 ±0.0823 | 0.6607 ±0.0785 | 0.6962 ±0.0862 |
| SLOTS(end-to-end) | **0.6955 ±0.1102** | **0.6952 ±0.0962** | **0.5999 ±0.1084** | **0.5819 ±0.1024** | **0.7105 ±0.0984** | **0.7600 ±0.0893** |

| Models | HAR | | | | | |
|---|---|---|---|---|---|---|
| | Accuracy | Precision | Recall | F1 score | AUROC | AUPRC |
| TS2Vec | 0.5738±0.0125 | 0.5555±0.0241 | 0.5768±0.0362 | 0.5565±0.0452 | 0.7623±0.0123 | 0.7413±0.0163 |
| TS2Vec++ | 0.6743 ±0.0698 | 0.6197 ±0.0541 | 0.5838 ±0.0751 | 0.5701 ±0.0832 | 0.8488 ±0.0564 | 0.8114 ±0.0452 |
| TS-TCC | 0.6596 ±0.0079 | 0.6154 ±0.0115 | 0.6158 ±0.0266 | 0.5459 ±0.0120 | 0.7511 ±0.0077 | 0.7414 ±0.0112 |
| TS-TCC++ | 0.6787 ±0.0128 | 0.6473 ±0.0093 | 0.6316 ±0.0096 | 0.6169 ±0.0005 | 0.7575 ±0.0053 | 0.7572 ±0.0159 |
| SimCLR | 0.6168 ±0.0347 | 0.6390 ±0.0478 | 0.5704 ±0.0563 | 0.5744 ±0.0698 | 0.7452 ±0.0452 | 0.7098 ±0.0365 |
| SimCLR++ | 0.6217 ±0.0657 | 0.6435 ±0.0593 | 0.6205 ±0.0741 | 0.5977 ±0.0863 | 0.9004 ±0.0634 | 0.7637 ±0.0413 |
| MixUp | 0.6277 ±0.0421 | 0.6393 ±0.0325 | 0.6138 ±0.0415 | 0.6055 ±0.0365 | 0.7705 ±0.0214 | 0.7481 ±0.0259 |
| MixUp++ | 0.6699 ±0.0358 | 0.6577 ±0.0563 | 0.6541 ±0.0684 | 0.6172 ±0.0687 | 0.8878 ±0.0632 | 0.8254 ±0.0412 |
| TFC | 0.6502 ±0.0658 | 0.6102 ±0.0753 | 0.6083 ±0.0861 | 0.6095 ±0.0742 | 0.7315 ±0.0436 | 0.7371 ±0.0438 |
| TFC++ | 0.7263 ±0.0362 | 0.6479 ±0.0214 | 0.6634 ±0.0421 | 0.6512 ±0.0369 | 0.8519 ±0.0128 | 0.8064 ±0.0231 |
| SLOTS(two-stage) | 0.6654 ±0.0896 | 0.6538 ±0.0968 | 0.6276 ±0.0785 | 0.6076 ±0.0993 | 0.8011 ±0.0741 | 0.7565 ±0.0635 |
| SLOTS(end-to-end) | **0.7312 ±0.0936** | **0.6661 ±0.0967** | **0.6720 ±0.0786** | **0.6615 ±0.1036** | **0.9013 ±0.0997** | **0.8811 ±0.0974** |

Table 6: Performance of baseline and SLOTS: pre-training on DEAP and fine-tuning on SEED.

| Models | Accuracy | Precision | Recall | F1 Score | AUROC | AUPRC |
|---|---|---|---|---|---|---|
| TS2Vec | 0.5781±0.0281 | 0.5683±0.0272 | 0.5325±0.0351 | 0.5278±0.0315 | 0.7488±0.0193 | 0.6151±0.0252 |
| TS-TCC | 0.4937±0.0250 | 0.5516±0.0147 | 0.4870±0.0275 | 0.4859±0.0202 | 0.6915±0.0332 | 0.5674±0.0101 |
| SimCLR | 0.5312±0.0312 | 0.5472±0.0363 | 0.5248±0.0284 | 0.5154±0.0252 | 0.7363±0.0173 | 0.6206±0.0159 |
| MixUp | 0.4875±0.0437 | 0.4832±0.0377 | 0.4878±0.0275 | 0.4647±0.0390 | 0.6887±0.0382 | 0.5677±0.0332 |
| TFC | 0.6031±0.0093 | 0.6135±0.0315 | 0.6028±0.0239 | 0.5847±0.0272 | 0.7900±0.0020 | 0.6642±0.0206 |
| SLOTS (two-stage) | 0.5250± 0.0062 | 0.5313±0.0185 | 0.5215±0.0112 | 0.5180±0.0122 | 0.7176±0.0050 | 0.5928±0.0014 |

APPENDIX REFERENCES

[S1] Sara Bagherzadeh, Keivan Maghooli, Ahmad Shalbaf, and Arash Maghsoudi. Emotion recognition using effective connectivity and pre-trained convolutional neural networks in EEG signals. *Cognitive Neurodynamics*, 16:1087–1106, 2022.

[S2] Hao Chao and Liang Dong. Emotion recognition using three-dimensional feature and convolutional neural network from multichannel EEG signals. *IEEE Sensors Journal*, 21(2):2024–2034, 2021.

[S3] Arvid Frydenlund and Frank Rudzicz. Emotional affect estimation using video and EEG data in deep neural networks. In *Canadian Conference on Artificial Intelligence*, pp. 273–280, 2015.

[S4] Byung Hyung Kim and Sungho Jo. Deep physiological affect network for the recognition of human emotions. *IEEE Transactions on Affective Computing*, 11(2):230–243, 2020.

[S5] Nandini Kumari, Shamama Anwar, and Vandana Bhattacharjee. Time series-dependent feature of EEG signals for improved visually evoked emotion classification using emotioncapsnet. *Neural Computing and Applications*, 34:13291–13303, 2022.

[S6] Zirui Lan, Olga Sourina, Lipo Wang, and Yisi Liu. Real-time EEG-based emotion monitoring using stable features. *The Visual Computer*, 32:347–358, 2016.

[S7] Zhen Liang, Rushuang Zhou, Li Zhang, Linling Li, Gan Huang, Zhiguo Zhang, and Shin Ishii. EEGFuseNet: Hybrid unsupervised deep feature characterization and fusion for high-dimensional EEG with an application to emotion recognition. *IEEE Transactions on Neural Systems and Rehabilitation Engineering*, 29:1913–1925, 2021.

[S8] Ahmet Mert and Aydin Akan. Emotion recognition from EEG signals by using multivariate empirical mode decomposition. *Pattern Analysis and Applications*, 21:81–89, 2018.

[S9] Tien Pham, Wanli Ma, Dat Tran, Duc Su Tran, and Dinh Phung. A study on the stability of EEG signals for user authentication. In *International IEEE/EMBS Conference on Neural Engineering*, pp. 122–125, 2015.

[S10] James A Russell. A circumplex model of affect. *Journal of Personality and Social Psychology*, 39(6):1161–1178, 1980.

