# OpenReview forum: "Semi-Supervised End-To-End Contrastive Learning For Time Series Classification"
_ICLR.cc/2024/Conference — ICLR 2024 Conference Withdrawn Submission_

### Official Review · Reviewer_DqhR · 2023-10-21

**Soundness:** 3 good
**Presentation:** 3 good
**Contribution:** 2 fair
**Rating:** 5
**Confidence:** 4

**Summary:**

This paper proposes a semi-supervised contrastive learning approach for time series classification tasks. The main contribution of the model is the proposed hybrid loss, which is a weighted sum of a self-supervised contrastive loss, a supervised contrastive loss, and a classification loss. Experiments on multiple time series classification datasets suggest that the proposed method is superior to prior two-stage contrastive learning methods.

**Strengths:**

1. The paper is easy to understand.
2. The methods are sound and the experimental designs are reasonable.
3. Strong empirical results compared to prior two-stage contrastive learning methods.

**Weaknesses:**

1. Technical contribution is marginal. Both self-supervised and supervised contrastive losses (Equations 4-5) exist in the literature and have been well studies. The hybrid loss was simply a weighted sum of the three losses. The backbone encoder and temporal masking augmentation was from the literature too.
2. No details about data split and hyperparameter selection.
3. Comparisons to baselines are not fair. Default hyperparameters were used for baselines. However, some baselines such as SimCLR was introduced in computer vision domain, and thus using default hyperparameters on time series data would result in suboptimal performance for the baselines. If hyperparameters for SLOTs were tuned, then hyperparameters for baselines should be tuned on the same data too.
4. Citation format is wrong, which reduces the readability of the paper.

**Questions:**

1. In SimCLR, a large number of negative pairs is needed. How many negative pairs were used in the self-supervised contrastive loss for SLOTS?
2. In ablation studies (Table 3), it would be better to see individual effects of supervised contrastive loss and classification loss.
3. How was data split done? How were hyperparameters selected?
4. If hyperparameters for SLOTS were optimized, please optimize hyperparameters for baselines for fair comparison.
5. How were baseline end-to-end models trained? Please clarify in "Baselines" section. Same question applies to two-stage version of SLOTs.
6. Please correct the citation format.

---

### Official Review · Reviewer_9UWV · 2023-10-31

**Soundness:** 2 fair
**Presentation:** 3 good
**Contribution:** 2 fair
**Rating:** 3
**Confidence:** 5

**Summary:**

The paper proposes a new method for semi-supervised learning of time-series. The method is based on contrastive learning for both supervised and unsupervised loss terms. The method uses DEAP, SEED, HAR, EPILEPCY, and P19 to evaluate the method.

**Strengths:**

The paper has several strengths:

- The paper targets an important and often neglected area in time-series representation learning.

- The paper is well-written and easy to follow.

- 5 datasets are used to evaluate its performance across different fields.

- Strong results are obtained.

**Weaknesses:**

The paper has several weaknesses:

1- The most important weakness is that the notion of using contrastive loss for semi-supervised leaning is not new. The approach has been used in a variety of semi-supervised literature. Examples include "Class-aware contrastive semi-supervised learning" (already cited in the paper), "CoMatch: Semi-supervised Learning with Contrastive Graph Regularization" (not cited), "Contrastive Regularization for Semi-Supervised Learning" (not cited), and others. In fact, the ablation study shows that almost all the contribution is coming from one of the losses (Ls) and the other isn't doing much. Examples of the general idea being used before is "Supervision Accelerates Pre-training in Contrastive Semi-Supervised Learning of Visual Representations".

2- One of the emphases of the paper seems to be the "end-to-end" aspect of the work. First, this is not unique to this paper, many other semi-supervised methods take a similar approach. Second, why is being end-to-end so important? The claim is that it is more efficient, but there are no studies to back this up. Moreover, while this may impact training, it doesn't seem to impact inference (which is what we really care about).

3- There are key papers that the paper does not compare its results to. Examples include MixMatch, FixMatch, AdaMatch, and others, which have been proposed in other areas (not necessarily for time-series). And then specifically for time-series, there are methods that are missing, e.g., "PARSE: Pairwise Alignment of Representations in Semi-Supervised EEG Learning for Emotion Recognition". All these missing comparisons make it hard to understand how good the method is comparison to others.

**Questions:**

Please see my comments under weaknesses.

---

### Official Review · Reviewer_5BkY · 2023-11-01

**Soundness:** 3 good
**Presentation:** 2 fair
**Contribution:** 2 fair
**Rating:** 5
**Confidence:** 4

**Summary:**

The paper describes a semi-supervised learning algorithm for time-series classification. Unlike the mainstream two-stage methods, the proposed approach is to train the model in a single training stage with both supervised and unsupervised loss functions.  The proposed solution outperforms 5+ baselines on five different medial domain time series classification datasets.

**Strengths:**

- The proposed method borrows insights from other semi-supervised and unsupervised learning paper from computer vision domain. Applies the same principles for the time series classification tasks. The proposed method is technically sound and well-described.
- The experiments are performed 5+ baselines with 5 medical domain time series classification (public) datasets. The paper also creates additional baselines by adapting the baselines for one-stage semi-supervised training.
- The proposed method outperforms all baselines on all datasets with a significant margin.

**Weaknesses:**

- The concept of one-stage semi-supervised learning is not novel in the field.
- The evaluations are limited to medical domain datasets. It is not clear how this method would perform for other domains.
- The authors suggest that the proposed approach is best suited for optimizing classification performance for target datasets, however, the learned representations would not necessarily transfer for addressing out of domain classification tasks.
- Robustness. There are missing details such as how the batching is performed, details of how the weights are determined in the loss function etc. It is not clear if the method requires rigorous hyper-parameter tuning or works out of the box with default parameters.
- The paper is missing a pseudo-labeling semi-supervised learning baseline. They are widely studied and commonly practiced in the applied settings due to the simplicity and ease of use.

**Questions:**

There are number of unknowns about the robustness / generalizability of the proposed method.
- Q1: How would the proposed method perform beyond medical datasets?
- Q2: How are the weights in the loss function determined? Do we have to determine new weights every time the labeled data ratio or dataset size changes? How about training schedule and learning rates? Any sensitivity?
- Q3: How robust is the proposed method against label noise?
- Q4: There are not sufficient details about how the batches are created and what happens to the supervised loss terms when the label is missing for the current training example during training.
- Q5: How does model capacity impact the training steps? Any known limitations in terms of underfitting or overfitting for certain scale of datasets and/or model sizes?

---

### Official Review · Reviewer_UUpX · 2023-11-06

**Soundness:** 2 fair
**Presentation:** 3 good
**Contribution:** 2 fair
**Rating:** 3
**Confidence:** 5

**Summary:**

The paper addresses two main limitations of existing two-stage contrastive learning methods in time series classification, i.e., the disconnect between unsupervised pre-training and downstream fine-tuning tasks, and the failure to leverage the full potential of ground truth-guided classification loss.

To tackle these problems, the paper introduces SLOTS, an end-to-end semi-supervised learning model for time series classification that effectively utilizes both labeled and unlabeled data. SLOTS simultaneously calculates unsupervised and supervised contrastive losses along with classification loss, optimizing both the encoder and classifier in a single process.

The authors show that their approach simplifies the learning framework besides improving the classification performance.

**Strengths:**

The paper is clear and the flow of writing is good.

**Weaknesses:**

I have major concerns regarding the methodology, summarized as follows.

- The novelty of this work is quite limited. The paper just adds unsupervised and supervised contrastive losses along with classification loss in an end-to-end way. Nothing is different about the methodology, the loss function, or the augmentations.
- Table 3 shows that the unsupervised contrastive loss is ineffective. Its removal has barely affected the performance. This means that supervised losses (as expected) are the most important ones, which deteriorates the significance of the end-to-end training that mainly targeted including the unsupervised contrastive loss.
- The authors have completely ignored semi-supervised time series representation learning methods in the literature review and did not use any of them as baselines. Despite having the “++” variants of self-supervised methods, this is just an added cross-entropy loss, and might not be the best way to get a good performance. Other methods proposed for semi-supervised time series can have better approaches to handling the data in a way that improves the performance, and therefore, should be included. Examples are:
[1] "Self-supervised learning for semi-supervised time series classification."*PAKDD*, 2020.
[2] "Self-supervised contrastive representation learning for semi-supervised time-series classification." *IEEE TPAMI* 2023.
[3]"Deep Semi-supervised Learning for Time-Series Classification." *Deep Learning Applications*, 2022.
[4] "Selfmatch: Robust semisupervised time-series classification with self-distillation", International Journal of Intelligent Systems 2022.
[5] "Semi-supervised time series classification by temporal relation prediction", ICASSP 2021.
- The hybrid loss is a weighted sum of the three losses. How did you calculate (or assign) these weights?
- The performance gap in Table 1 is significantly high and not well-justified, especially with the smaller fractions of data. Also, I find this an issue with some of the results. Specifically, I noticed that SLOTS is basically the same as SimCLR++, but with a supervised contrastive loss (Ls). However, The accuracy of SimCLR++ in Table 1 is 0.5862, while SLOTS (w/o Ls) in Table 3 is 0.6103.
- In the experiments, the settings of the “two-stage SLOTS” are not clear. Do you use the supervised contrastive loss in fine-tuning or not?
- In the ablation study, I expected to see the performance without both contrastive losses, i.e., only cross-entropy loss, to see how the addition of contrastive learning was useful.
- Since you attempt to achieve a fair comparison in your experiments, I noticed that most of the baselines are using 1D convolutions, while you use a 2D convolution. This can a minor issue, but it can affect the comparison.
- What does “Full model” mean in Table 3 in SLOTS (two-stage)?

**Questions:**

I expect the authors to answer the above weakness points.

---

### Official Review · Reviewer_F7FV · 2023-11-07

**Soundness:** 3 good
**Presentation:** 3 good
**Contribution:** 3 good
**Rating:** 6
**Confidence:** 4

**Summary:**

This paper proposes a new end-to-end semi-supervised method for contrastive models where the model is fed both unlabeled and labeled data with 3 different losses that take into account where its sample comes from and optimize both unsupervised, supervised, and classification objectives.

**Strengths:**

- Strong results in a wide array of tasks
- Intuitive explanation of the method

**Weaknesses:**

- There are no semi-supervised baselines

**Questions:**

- Can the authors comment on the lack of semi-supervised baselines? All baselines seem to be contrastive models repurposed to perform semi-supervised learning within the proposed framework.